# Isolation and Characterization of a Novel *Aeromonas salmonicida*-Infecting *Studiervirinae* Bacteriophage, JELG-KS1

**DOI:** 10.3390/microorganisms12030542

**Published:** 2024-03-08

**Authors:** Karina Svanberga, Jelena Avsejenko, Juris Jansons, Davids Fridmanis, Tatjana Kazaka, Aivars Berzins, Andris Dislers, Andris Kazaks, Nikita Zrelovs

**Affiliations:** 1Latvian Biomedical Research and Study Centre, Ratsupites 1 k-1, LV-1067 Riga, Latvia; karina.svanberga@biomed.lu.lv (K.S.); jansons@biomed.lu.lv (J.J.); davids@biomed.lu.lv (D.F.); tatyanav@biomed.lu.lv (T.K.); dishlers@biomed.lu.lv (A.D.); andris@biomed.lu.lv (A.K.); 2Institute of Food Safety, Animal Health and Environment “BIOR”, Lejupes 3, LV-1076 Riga, Latvia; jelena.avsejenko@bior.lv (J.A.); aivars.berzins@bior.lv (A.B.)

**Keywords:** bacteriophage, whole-genome sequencing, genomics, aquaculture, *Aeromonas salmonicida*

## Abstract

Representatives of the bacterial genus *Aeromonas* are some of the most notorious aquaculture pathogens associated with a range of diseases in different fish species. As the world forges toward the post-antibiotic era, alternative options for combating bacterial pathogens are needed. One such alternative option is phage biocontrol. In this study, a novel podophage—JELG-KS1—infecting *Aeromonas salmonicida* was retrieved from wastewater along with its host strain. The genome of the JELG-KS1 phage is a 40,505 bp dsDNA molecule with a GC% of 53.42% and 185 bp direct terminal repeats and encodes 53 predicted proteins. Genomic analysis indicates that JELG-KS1 might represent a novel genus within the subfamily *Studiervirinae*. Podophage JELG-KS1 is a strictly lytic phage without any identifiable virulence or AMR genes that quickly adsorbs onto the surface of host cells to initiate a 48 min long infectious cycle, resulting in the release of 71 ± 12 JELG-KS1 progeny virions per infected cell. JELG-KS1 effectively lyses its host population in vitro, even at very low multiplicities of infection. However, when challenged against a panel of *Aeromonas* spp. strains associated with diseases in aquaculture, JELG-KS1 shows host-specificity that is confined only to its isolation strain, immediately compromising its potential for *Aeromonas* spp. biocontrol in aquaculture.

## 1. Introduction

Aquaculture is currently one of the fastest-growing food industry sectors, and its development is powered by both the expansion and intensification of production [1,2,3]. Intensification, however, is not only causing stress in fish, which depresses their immunity, but is also often approached without adequate attention to the maintenance of water quality, general hygiene, and biosecurity measures [4,5,6,7]. This makes aquaculture one of the sectors especially prone to bacterial disease occurrence, which is being overcome by the rather reckless use of antibiotics, making aquaculture, especially fish farming, not only one of the sectors that especially suffers from but also further propagates, antibiotic resistance [8,9,10].

Among the numerous bacterial pathogens endangering aquaculture, *Aeromonas* spp. are considered one of the most worrisome pathogens responsible for the significant mortality of fish, leading to severe economic losses worldwide [11]. Representatives of the *Aeromonas* genus are Gram-negative rods commonly found in, but not limited to, various aquatic environments. Their environmental versatility, coupled with the ability of some strains to produce a number of virulence factors (e.g., proteases, hemolysins, and cytotoxins), enables them to colonize and infect a variety of host species [12]. Based on their growth and biochemical characteristics, *Aeromonas* spp. were historically divided into two main groups—non-motile psychrophiles (with an optimal growth temperature of around 22 °C) and mesophiles having a single polar flagellum (that grow well at 35–37 °C) [13,14]. Although *Aeromonas* spp. are mainly thought to be associated with a range of fish diseases (e.g., furunculosis and hemorrhagic septicemia), reports of human infections by some *Aeromonas* species are also not infrequent [15,16].

Of the various *Aeromonas* species, *A. salmonicida* is often pinpointed as one of the main fish pathogens of concern due to its ability to cause furunculosis, which affects a wide range of fish species, especially salmonids (e.g., Atlantic salmon, rainbow trout, etc.) [17]. *A. salmonicida* can further be classified into five recognized subspecies comprising traditionally psychrophilic (subsp. *salmonicida*, *achromogenes*, and *smithia*), mesophilic (subsp. *pectinolytica*), as well as intermediary lifestyle (subsp. *masoucida*) strains [18]. Interestingly, accumulated evidence suggests that *A. salmonicida* subsp. *salmonicida* encompasses both mesophilic and psychrophilic bacterial strains [19]. Infections caused by *A. salmonicida* in aquaculture were previously treated using antibiotics such as oxytetracycline, florfenicol, flumequine, and enrofloxacin administered through injections or feed, but the emergence of antibiotic-resistant strains and the long-term environmental risks associated with excessive antibiotic usage compromise their efficacy and prompt the use of other options for *A. salmonicida* control in aquaculture [20,21,22,23]. Such proposed alternative options include, but are not limited to, the selective breeding of heritable resistance, fish vaccination, and the usage of prebiotics and probiotics as immunostimulants, among others, each with its own pros and cons [24,25,26,27,28,29]. Yet, the associated protection has been inconsistent, both within and among different species of fish, and the nature of this protection is still not completely clear in many cases.

Recently, another alternative option—phage therapy—for the (bio)control of pathogens of bacterial origin has seen renewed interest [30]. Bacteriophages, which are viruses of bacteria, are abundant in every ecological niche, where they serve as natural predators of bacteria, and their target specificity, coupled with the ability of most virulent phages to efficiently self-propagate upon contact with the susceptible host population, makes them a promising option for eliminating unwanted bacteria without greatly disturbing the other microbiota in a given environment [31]. The aquaculture setting seems especially welcoming to the use of bacteriophages due to their ability to spread via passive diffusion and to eventually encounter a susceptible host [32]. Thus, numerous recent studies have described the isolation and characterization of phages targeting bacterial fish pathogens, as well as shown phage efficacy in decreasing the symptoms of associated diseases and reducing fish mortality in laboratory-scale trials [33]. Moreover, several phage-based biocontrol solutions for bacterial pathogens in aquaculture are currently either available or being developed [34]. However, the host specificity of phages is both a blessing and a curse, and the development of phage-based solutions that would cover a sufficiently broad potential target range, given that some phages might be specific only to several particular strains of a pathogen species of interest, is not only time-consuming but also requires a thorough characterization of the prospective phage candidates, which might reveal their inapplicability for therapeutical/biocontrol purposes at any stage during the study of their properties [35,36].

As of the 8th of February 2024, GenBank contained genomes of 178 phages designated as *Aeromonas* phages (Appendix A) [37]. These genomes ranged from 30,791 bp (phages ZPAH7 and ZPAH7B) to 262,372 bp in length (phage D3) and had a GC content of 34.43% (phage vB_AdhM_DL) to 62.07% (phage AhMtk13b), while being scattered among at least eight viral families based on the taxonomy associated with the submissions. *Aeromonas hydrophila* was specifically indicated as a host for 86 of the phages, followed by 44 *Aeromonas salmonicida*-specific phages, the remaining phages either infected other *Aeromonas* species (17/178) or did not have a particular host species indicated at all (31/178).

In this study, a novel lytic bacteriophage, JELG-KS1, infecting a strain of *Aeromonas salmonicida* retrieved from wastewater is reported and thoroughly characterized. Transmission electron microscopy revealed that JELG-KS1 virions have a podophage morphotype (icosahedral capsid with a short non-contractile tail). Based on its complete genome sequence, *Aeromonas* phage JELG-KS1 might be considered a founding member for a tentative novel phage genus within the family *Autographiviridae*, subfamily *Studiervirinae*. Although the properties of phage JELG-KS1 demonstrated on its isolation host tentatively made it a promising “workhorse” for potential practical applications, it failed to lyse any of the 19 other *Aeromonas* spp. strains previously isolated from aquaculture environments in Latvia, making it somewhat of a “one-trick pony”. This study further highlights the necessity to always challenge the newly isolated phages against a panel of different relevant hosts before proposing any phage as a prospective constituent agent for the development of phage-mediated biocontrol preparations.

## 2. Materials and Methods

### 2.1. Host Isolation and Identification

Wastewater samples (1 L) were collected from the incoming water flow of the Jelgava city wastewater treatment plant SIA “JELGAVAS ŪDENS” on 13 April 2021 and stored at 4 °C for a few days until a 1 mL aliquot was taken and diluted 100× in 0.8% NaCl solution. Aliquots (50 µL) of the diluted sample were spread onto four Petri dishes with solidified “Peptone Mix” media (Peptone Mix (g/L): peptone—10 (Fluka, Buchs, Switzerland), yeast extract—5 (Sigma-Aldrich, St. Louis, MO, USA), NaCl—5 (Sigma-Aldrich), Bacto agar—15 (Sigma-Aldrich)). Two of the dishes were incubated overnight at room temperature (RT) and the other two at 37 °C. 

Individual morphologically distinct bacterial colonies were purified by subculturing twice. After subculturing, a few colonies of the isolate with the working name “JK” were suspended in 300 µL 0.8% NaCl, and 1 µL of proteinase K (18.7 mg/mL; Thermo Fisher Scientific, Waltham, MA, USA) and SDS (Sigma-Aldrich, 0.5% final concentration) were added to the suspension. The suspension was then briefly vortexed and incubated at 56 °C for 1 h before being subjected to DNA extraction using the Genomic DNA Clean & Concentrator-10 Kit (Zymo Research, Irvine, CA, USA) according to the manufacturer’s guidelines. In parallel, 5 mL of liquid LB medium (Lysogeny broth (g/L): tryptone—10 (Sigma-Aldrich), yeast extract—5 (Fluka), NaCl—10) was seeded with an isolated JK colony and stationarily incubated at RT overnight to obtain a liquid indicator culture for phage hunting.

To amplify the 16S rRNA gene region of interest, a PCR (program: 3 min at 95 °C, 35 cycles of (30 s at 95 °C, 30 s at 55 °C, 1.5 min at 72 °C), 5 min at 72 °C, followed by a hold at 4 °C) using universal 27F and 1492R primers (ordered from Metabion, Steinkirchen, Germany) was carried out using ~100 ng of DNA of “JK” as a template [38]. The PCR product was then subjected to native agarose gel electrophoresis. The gel was then visualized under UV illumination, and the band corresponding to the amplified partial 16S rRNA gene sequence of JK (~1450 bp) was excised from the gel and further extracted using the GeneJET Gel Extraction Kit (Thermo Fisher Scientific) according to the manufacturer’s protocol, performing the additional optional step for further sequencing of the extracted DNA.

Sanger-based sequencing [39] of the extracted PCR product was performed as two separate reactions to obtain forward and reverse reads (from the 27F and 1492R primers, respectively) using ABI PRISM 3130xl (Thermo Fisher Scientific) as a sequencer. Reactions were prepared according to the recommendations of the BigDye^®^ Terminator v3.1 Cycle Sequencing Kit (Applied Biosystems, Waltham, MA, USA). Obtained read chromatograms were inspected in GeneStudio (v2.2.0.0.), and the reads had their low-confidence terminal traces trimmed before being assembled into a contig based on their overlapping regions (higher-quality traces from either read were used for the consensus sequence of the overlap). 

The resulting near-complete 16S rRNA gene sequence of isolate JK was queried against the EzBioCloud database [40] to determine the most closely related bacterial species.

### 2.2. Phage Isolation, Propagation, and Purification

To isolate phages able to infect isolate JK, 15 mL of the same wastewater sample was centrifuged for 30 min at 4629× *g* in an Eppendorf 5810R centrifuge (Eppendorf, Hamburg, Germany), and the supernatant was decanted and filtered through a 0.45 µm pore size syringe filter (Sarstedt, Nümbrecht, Germany). The filtrate was next concentrated on an Amicon Ultra-15 100 kDa MWCO filter (Merck, Kenilworth, NJ, USA) at 3214× *g* in an Eppendorf 5810R centrifuge to a final volume of 0.5 mL. Afterward, 100 µL of the filtrate was mixed with 50 µL of the previously obtained indicator JK culture, and ~7 mL of the melted Luria top-layer agar (Luria 0.7% (g/L): tryptone—10 (Sigma-Aldrich), yeast extract—5 (Fluka), NaCl—0.5, Bacto Agar—7 (Sigma-Aldrich)) was spread onto a Petri dish with solidified Peptone Mix as the bottom-layer (1.5%) agar; this was repeated for several Petri dishes. The Petri dishes were then incubated both at 37 °C and at RT overnight. After incubation, several seemingly morphologically different cell-lysis zones (assumed to be phage plaques) were independently collected from the top agar layer and transferred to 1.5 mL Eppendorf tubes containing 1 mL of 0.8% NaCl solution. Plaques were then purified by successively repeating the same double-agar overlay procedure three times using JK as the indicator culture for the host lawn and phage plaques from the previous passage. After this step, it was concluded that, based on the coinciding plaque morphologies, all plaque lines likely represent a single individual phage, which we named “JELG-KS1”.

For JELG-KS1 propagation, one of the plaques from the final passage was suspended in 1 mL of 0.8% NaCl, and 100 µL was added to a test tube with 5 mL of exponentially growing culture of the JK bacterial isolate in LB liquid medium. The infected culture was then left to incubate in an Infors multitron incubator shaker (Infors AG, Bottmingen, Switzerland) overnight (80 rpm, 37 °C). The next day, the lysate was subjected to centrifugation in an Eppendorf 5810R centrifuge at 4629× *g* for 10 min, and the supernatant was used to infect a flask containing 100 mL of LB medium with an exponentially growing JK culture. After being incubated overnight as described in the previous step, the lysate was clarified at 4629× *g* for 30 min using an Eppendorf 5810R centrifuge and filtered through a 0.45 µm pore-size syringe filter (Sarstedt, Nümbrecht, Germany). To concentrate the filtered phage lysate, ultracentrifugation for 1 h at 48,384× *g* at 4 °C was performed using the Optima TL-100 ultracentrifuge (Beckman Coulter, Brea, CA, USA). The supernatant was decanted, and 2 mL of 20 mM Tris buffer (pH 8) was used to dissolve the phage pellet.

The obtained suspension was then layered on top of a CsCl solution (CsCl—0.6 g/mL in TE buffer (*w*/*v*)); two tubes (Ultra-Clear centrifuge tubes, 14 × 95 mm, Beckman Coulter) were filled with 11.5 mL of the CsCl solution, and 1 mL of a concentrated phage sample was loaded on top of the CsCl solution. Tubes were centrifuged at 24,000 rpm (100,000× *g* max) at 15 °C for 20 h using an SW 40 Ti rotor (Beckman Coulter) in the Beckman Optima L-100XP ultracentrifuge. The distinct phage-containing band was collected by pipetting and was further desalted by washing twice with 5 mL phosphate-buffered saline (PBS, pH 7.4) using an Amicon Ultra-15 100 kDa MWCO filter (Merck) (centrifuged on Eppendorf 5810R for 7 min at 3214× *g*). The resulting sample was aliquoted; half of the aliquots proceeded to long-term storage at −80 °C, and the other half was stored at 4 °C and used for further experiments as needed, including subsequent propagation and purification as described above to obtain more JELG-KS1 phage.

### 2.3. Transmission Electron Microscopy and Virion Dimension Measurements

An aliquot of the CsCl-gradient-purified and desalted JELG-KS1 specimen (5 µL) was allowed to adsorb onto carbonized formvar-coated 300-mesh copper grids (Agar Scientific, Stansted, UK) for 5 min. Afterward, the sample was rinsed with 1 mL of 1 mM EDTA solution and negatively stained with 0.5% uranyl acetate solution for 1 min, then allowed to dry for a few hours before being examined under a JEM-1230 transmission electron microscope (JEOL, Akishima, Japan). JELG-KS1 virion micrographs were taken with a Morada 11 MegaPixel TEM CCD microscope-mounted camera (Olympus, Tokyo, Japan), and virion dimensions were measured using ImageJ 1.52v software [41] using a scale bar for a pixel-to-nm ratio. The virion capsid diameter and tail length of 10 randomly selected virions from micrographs taken at different fields of view were measured using the “straight line” utility; the provided dimensions represent the average ± standard deviation.

### 2.4. JELG-KS1 Adsorption Test

The phage adsorption test was performed according to the method described by [42], with some modifications. A phage sample (100 µL, ~2 × 10^7^ PFU/mL) was added to a flask containing 9.9 mL of LB medium with an exponentially growing host culture (~2 × 10^8^ CFU/mL) for a multiplicity of infection (MOI) of ~0.001. The flask was incubated at RT and periodically swirled by hand. Throughout the incubation, 100 µL aliquots were collected at 1, 2, 3, 4, 5, 10, and 20 min post-infection, immediately mixed with 900 µL of LB liquid medium, and placed on ice before being centrifuged at 12,000× *g* for 2 min, followed by the preparation of serial dilutions of the supernatant. The number of unadsorbed virions in the supernatant at the given time point after the infection was determined in triplicate using the double-agar overlay method. After incubation of the double-agar overlay plates at 37 °C for ~20 h and determination of the corresponding PFU/mL values, obtained titers were normalized against the PFU/mL at the point of infection. The results were averaged across five independent experiments.

### 2.5. One-Step Growth of JELG-KS1

A one-step growth experiment for phage JELG-KS1 was performed according to the protocol described by Kropinski [43], with several modifications. Briefly, 100 µL of phage JELG-KS1 lysate (~1 × 10^7^ PFU/mL) was added to 9.9 mL of the exponentially growing isolate JK culture (~1 × 10^8^ CFU/mL) at a multiplicity of infection (MOI) of ~0.001 (without the addition of CaCl_2_). The mixture was then incubated at RT with periodical swirling by hand throughout the entire experiment. Aliquots were taken at various time points after the infection. Rapid centrifugation (12,000× *g*, 2 min) was performed to separate phage-infected cells from the free phage particles in the adsorption flask instead of using chloroform for unadsorbed virion detection as described in the protocol [43]. The experiment was repeated three times.

### 2.6. JELG-KS1 In Vitro Lysis Profiles

For the in vitro lysis assay, 190 µL of LB medium containing host culture at the early exponential growth phase (~1 × 10^8^ CFU/mL) was mixed with 10 µL of phage lysate samples pre-diluted to result in MOIs of 2, 0.2, 0.02, 0.002, 0.0002. An uninfected bacterial culture (190 µL) with the addition of 10 µL of sterile LB medium served as a negative control. Sterile LB medium (200 µL) served as a relevant background sample. Each condition was replicated in 8 different wells of a 96-well plate; a multichannel pipette was used to ensure synchronous infections per each MOI tested. Optical density at a 595 nm wavelength was measured using a VICTOR3V microplate reader (PerkinElmer, Waltham, MA, USA) at RT (~23 °C in the room with the microplate reader during the experiments). Measurement of each sample-containing well was repeated with an interval of 5 min for a total duration of more than 5 h after the infection.

### 2.7. JELG-KS1 Thermal and pH Stability

Prior to JELG-KS1 virion temperature stability assays, the JELG-KS1 lysate (~2 × 10^9^ PFU/mL) was diluted 100-fold in 0.8% NaCl solution. Afterward, 1 mL aliquots of phage-containing 0.8% NaCl solution (~ 2 × 10^7^ PFU) were transferred to 1.5 mL tubes (Eppendorf) and mixed, then incubated in a TDB-120 thermoblock (Biosan, Riga, Latvia) at different temperatures (25, 35, 40, 45, 50, and 55 °C) for 1 h. After the hour-long incubation, 5 µL aliquots of serial dilutions of the samples subjected to different temperatures were spotted onto freshly solidified double-agar overlay plates with the soft-layer agar seeded with the JK culture for the lawn.

To test JELG-KS1 virion stability in different acidic and alkaline conditions, phage lysate at ~2 × 10^9^ PFU/mL was first diluted 10x in 0.8% NaCl, and then 100 µL aliquots were transferred to 1.5 mL tubes (Eppendorf) containing 900 µL of PBS preadjusted with 6 M HCl and 6 M NaOH to yield pH values of 3, 4, 5, 6, 7, 8, 9, 10, 11, and 12 and mixed. Samples were then left on the bench for 1 h at RT. Afterward, 5 µL aliquots of serial dilutions of the samples subjected to different pH conditions for an hour were spotted onto solidified double-agar overlay plates with the host bacteria mixed in for a lawn in the soft-layer agar. 

Double-agar overlay plates with the phage added were next incubated at 37 °C for ~20 h, and the resulting phage plaques were counted, allowing the determination of the PFU/mL remaining in the sample subjected to each tested condition (either temperature or pH). Results of three independent experiments performed within a few days were aggregated for both thermal and pH stability, respectively. To avoid the limit of detection possibly concealing the real PFU/mL count under extreme temperature and pH conditions imposed by spotting small volumes of the sample (5 µL per spot), whole undiluted volumes of the tubes incubated either at 55 °C, pH 3, or pH 12 were plated using the standard double-agar overlay procedure (up to 250 µL per plate) instead of spot testing.

To determine whether there are any statistically significant differences between the remaining PFU/mL count after incubation of the phage solutions for an hour at 25 °C or pH 7, which served as the respective temperature and pH stability controls, and other temperature or pH conditions within a given experiment, the Wilcoxon rank–sum test [44,45] was performed with Benjamini–Hochberg [46] correction to adjust the *p*-values for multiple comparisons within both thermal and pH stability testing using the “compare_means” function from the ggpubr R package (v0.6.0 [47]). Adjusted *p*-values of less than 0.05 were considered statistically significant (α = 0.05). The obtained PFU/mL counts were next transformed into log_10_(PFU/mL + 1) for visualization using the ggplot2 R package (v3.4.2 [48]).

### 2.8. JELG-KS1 Host-Range Determination

To test the ability of phage JELG-KS1 to lyse *Aeromonas* isolates relevant to aquaculture, 19 isolates identified as *Aeromonas* spp. that were previously isolated from Latvian aquaculture settings (diseased fish) were obtained from the bacterial strain collection of the Institute of Food Safety, Animal Health, and Environment “BIOR” (Table 1). 

Individual colonies of the respective isolates were picked from the Petri dishes containing blood agar (blood agar (g/L): tryptose—10, beef extract—10, sodium chloride (NaCl)—5, agar—15 (BioLife, Milan, Italy)) with the addition of 5% defibrinated horse blood (TCS Biosciences Ltd., Botolph Claydon, UK) received from BIOR was used to seed 5 mL of LB liquid medium. After overnight incubation at room temperature, the respective cultures were used as a lawn for double-agar overlays (bottom-layer agar—1.5% Peptone Mix; top-layer agar—0.7% LB), and spot tests using 5–50 µL of the phage JELG-KS1 lysate at ~2 × 10^9^ PFU/mL. The plates with each of the bacterial strains used for the lawn were incubated both at RT and 37 °C for up to several days.

The *Aeromonas* spp. identity of bacterial isolates was first verified with 16S rRNA gene sequencing using 27F and 1492R primers as described for the identification of the isolation host of phage JELG-KS1 in the corresponding Materials and Methods section.

To approximate the relationships between the isolation host of JELG-KS1, other *Aeromonas* isolates used for host-range screening, as well as related bacteria for context, a partial 16S rRNA gene sequence phylogeny was reconstructed. All validly named EzBioCloud hits (n = 45) to the respective sequence of the JELG-KS1 isolation host, JK, were selected and downloaded to serve as a context to visualize the place of the JK isolation host and the other strains that JELG-KS1 was tested against (n = 19) within the current taxonomical framework of related bacteria. Multiple-sequence alignment (MSA) was performed using MAFFT (v7.453; [37]). The obtained MSA (n = 65) was then trimmed using gblocks (v0.91b; [49]) under default settings, and the trimmed MSA was used for the neighbor-joining-tree [50] reconstruction in MEGA7 (v7.0.26; [51]), including both transitions and transversions, assuming uniform rates among sites, and computing the evolutionary distances using the p-distance method. Branch supports were assessed using 1000 bootstrap test replicates [52]. The resulting tree was midpoint-rooted and visualized in FigTree (v1.4.4.; [53]; accessed on 10 May 2021).

As 16S rRNA sequence phylogenies are known to be inadequate in discerning between some of the currently recognized closely related *Aeromonas* species, the resolution of the available *Aeromonas* strain identification was improved by *gyrB* gene sequence analysis of the isolates [54]. To this end, custom primers that would bind within the *gyrB* gene of the *Aeromonas* spp. most closely related to our strains (as determined by the 16S rRNA gene phylogeny) were designed and ordered from Metabion (Table 2). A near-complete sequence of the *gyrB* gene was amplified from the genomic DNA of the respective strains using PCR; it was next sequenced using the Sanger-based method with subsequent scaffolding of the reads. In general, the same methodology as described in Section 2.1. for the 16S rRNA gene was followed. However, this time, primers Fw1_Aeromonas_gyrB and Rv2_Aeromonas_gyrB were used to amplify a ~2.3 kb region of *gyrB* gene, and 57 °C was set for the annealing steps of the PCR program. The Sanger-based sequencing of the amplified *gyrB* fragment was carried out in four separate reactions to obtain reads from each of the primers listed in Table 2. Corresponding *gyrB* reads were further assembled into contigs representing a near full-length *gyrB* gene sequence for each of the isolates, including the isolation host of JELG-KS1 (isolate JK). Multiple *gyrB* sequence alignments, as well as the subsequent NJ tree generation, were performed in the same way as for the 16S rRNA sequences. It was noted that not all the strains representing species related to strain JK that were used for phylogenetic context in the case of the 16S rRNA tree have their whole genome sequences or at least a *gyrB* gene sequenced and publicly available. The *gyrB* sequence of such strains was either substituted by the *gyrB* sequence originating from another related strain representing the same species (if possible) or omitted altogether from the performed *gyrB* sequence NJ tree reconstruction.

### 2.9. JELG-KS1 Genomic DNA Extraction and Whole-Genome Sequencing

Phage genomic DNA was extracted from 300 µL of a CsCl gradient-purified and desalted JELG-KS1 specimen employing the same protocol utilized for bacterial genomic DNA extraction (Section 2.1). The obtained DNA was evaluated qualitatively and quantitatively using the NanoDrop ND-1000 spectrophotometer (Thermo Scientific, Pittsburgh, PA, USA) and Qubit fluorometer (Invitrogen, Waltham, MA, USA) with a dsDNA high-sensitivity quantification assay (Invitrogen). Approximately 200 ng of JELG-KS1 dsDNA was subjected to random physical shearing via sonication in a Covaris S220 focused ultrasonicator (Covaris, Woburn, MA, USA) based on a protocol for a fragment length of 550 bp. Fragmented DNA was then used as input to prepare Illumina MiSeq-compatible DNA libraries for 250 bp paired-end read sequencing using the TruSeq DNA Nano Low-Throughput Library Prep Kit (Illumina, San Diego, CA, USA) protocol with adapter #5 from TruSeq DNA Single Indexes Set A (Illumina) to allow for further pooling with other unrelated libraries. The quality and quantity of the resulting library were inspected using an Agilent 2100 bioanalyzer (Agilent, Santa Clara, CA, USA) with a High Sensitivity DNA Kit (Agilent), as well as a Qubit fluorometer (Invitrogen) dsDNA high-sensitivity quantification assay (Invitrogen). Afterward, the JELG-KS1 genomic-DNA-containing library was pooled with 11 other unrelated and uniquely barcoded libraries and sequenced on the Illumina MiSeq system (Illumina) with the 500-cycle MiSeq Reagent Kit v2 nano (Illumina).

### 2.10. JELG-KS1 Genome De Novo Assembly and Validation

FastQC (v0.11.9; [55]) revealed that the demultiplexed raw-read dataset of the phage JELG-KS1 genome fragment library had 87,257 paired-end reads (up to 251 bp in length).

Putatively remaining adapter sequences were removed using the bbduk tool from the bbmap package [56], the trailing base of each read was trimmed off, and any reads shorter than 50 bp in length were discarded. De novo assembly was performed by Unicycler (v0.4.8; [57]) in “normal” mode, using a trimmed-read dataset as an input, which resulted in an assembly containing a contig representing the complete genome of JELG-KS1. 

In an attempt to correctly organize the assembled genome, PhageTerm (v1.0.12; [58]) was run on a 40,320 bp long contig flagged as “circular”, thus representing the complete genome of a novel phage, using untrimmed reads as input. Short direct terminal repeats of 185 bp were identified by PhageTerm, and the genome was reorganized accordingly. Raw reads were next mapped unto the PhageTerm-reorganized genome of JELG-KS1 using BWA-MEM (v0.7.17-r1188; [59]), with inspection of the sequence alignment map afterward for any coverage dips and assembly ambiguities in UGENE (v37.0; [60]). Mapping revealed that JELG-KS1 was sequenced to a mean genome depth of 468x and a whole genome coverage of ≥116x.

The presence of short direct terminal repeats was additionally validated by Sanger-based sequencing of the genomic JELG-KS1 DNA using custom primers designed to hybridize upstream of the predicted terminal-repeat sequences.

### 2.11. JELG-KS1 Genome Functional Annotation

DNA master (v5.23.6; https://phagesdb.org/DNAMaster/; accessed on 12 April 2022) was used as a sequence explorer to proceed with the genome functional-annotation process as described previously [61]. 

Briefly, open reading frame (ORF) predictions were carried out using Glimmer [62] and GeneMark [63] (ATG, GTG, CTG, and TTG were considered possible start codons; a >30 aa putative product length was required, with initial preference being given to the start codon resulting in the longest possible product; ORF overlaps of >100 bp were not allowed), whereas tRNA gene predictions were performed using ARAGORN [64] and tRNAscan [65]. The putative product of each ORF was then (in late April 2022) subjected to an NCBI conserved domain search (default settings, [66]), BLASTp analysis against a non-redundant protein sequence database (1 × 10^−3^ e-value cutoff, [67]), and an HHpred search (databases: PDB, Pfam, UniProt-SwissProt-viral70, CDD; default settings, [68]). Taking the results of the aforementioned searches into account, start codons of the predicted ORFs were evaluated and corrected where relevant. Afterward, the presence of putative Shine–Dalgarno (SD) sequences complementary to the antiSD sequence of *A. salmonicida* (13 bases of the *A. salmonicida* 16S rRNA tail; 3′-AUUCCUCCACUAG-5′ [69]) was inspected in the regions 20 bp upstream of the selected start codons for each ORF using free_align.pl script [70].

The complete annotated genome of *Aeromonas* phage JELG-KS1 was deposited into GenBank and is publicly available under accession number ON604651.

### 2.12. Relationships of JELG-KS1 to Other Phages

The complete genome sequence of *Aeromonas* phage JELG-KS1 was first subjected to a BLASTN search against the non-redundant NCBI Nucleotide database subset comprising sequences of viral origin (taxid:10239). Afterward, 35 top-scoring hits to genomes of other phages with a query coverage of ≥25% were downloaded and subjected to their pairwise intergenomic-similarity calculations using VIRIDIC (v1.0; [71]) under default settings. The resulting VIRIDIC intergenomic similarity matrix was then further manually annotated at the phage genus-level intergenomic similarity threshold (~70%) in Inkscape (v1.0.1; available online: https://inkscape.org; accessed on 10 May 2021).

A single representative per recognized or putative genera identified by VIRIDIC was selected for a more elaborate comparison of their genome architectures and contents in relation to JELG-KS1. For this, genomes of *Dickeya* phage Ninurta (NC_047964; *Ningirsuvirus*) and *Vibrio* phage ICP3 (MT740748; *Chatterjeevirus*), as well as *Aeromonas* phage T7-Ah (MT740748; yet without an official genus-level taxonomy) and *Escherichia* phage vB_Eco_Titus (OX090892; yet without an official genus-level taxonomy) were downloaded to be used for subsequent analyses. Product annotations seen in the submissions were retained; however, the vB_Eco_Titus genome was rearranged to begin with the original base 34,500 to ensure collinearity with other selected phages and JELG-KS1. To generate genome maps and comparisons of JELG-KS1 to the aforementioned evolutionarily related phages, Easyfig (v2.2.2; [72]) was used for nucleotide sequence comparisons using both BLASTn and tBLASTx modes, while Clinker (v0.0.23; [73]) was selected to compare the respective phage protein sequences. Shared genome-encoded product identification was also performed using Roary (v3.13.0; [74]) under different identity thresholds.

Several proteins (RNA polymerase, major capsid protein, DNA polymerase, and terminase large subunit) of JELG-KS1, for which pairwise functional independence in the phage lytic cycle was assumed, were selected to serve as marker proteins for phylogeny reconstructions. Corresponding protein amino acid sequences were queried against the non-redundant protein sequences of viral origin (taxid:10239) using BLASTp under default searching parameters, although the maximum number of target sequences was expanded to 500. For each selected protein, the top 25 highest-scoring hits were retrieved, and an outgroup was chosen randomly from the lower-scoring hits. The only outgroup selection criterion was that their bitscores were to be 70–80% of the bitscores from the least-scoring hits from the top 25 homologs. The dataset for each selected marker protein of JELG-KS1 was then aligned using MAFFT (v7.453; [75]) and subjected to maximum-likelihood tree reconstruction using IQ-TREE (v2.0.6; [76]). For each IQ-TREE run, the best-fit substitution model was selected using ModelFinder [77], polytomies were allowed, and 1000 ultrafast bootstrap (UFBoot; [78]) replicates were used as a measure of branch support. The resulting trees were outgroup-rooted and visualized in FigTree (v1.4.4; [53]; accessed on 10 May 2021).

Additionally, vConTACT2 (v0.11.3; [79]) was used under default settings using the relevant INPHARED [80] 1 October 2022 release inputs (representing 18,553 completely sequenced phages, including JELG-KS1). The vConTACT2-generated protein-sharing network was visualized and annotated in Cytoscape (v3.8.2; [81]). JELG-KS1 and its first neighbors were detached to a new network, which was further visualized using an edge-weighted spring-embedded layout and annotated.

## 3. Results and Discussion

### 3.1. Plaque and Virion Morphology of JELG-KS1 Phage

On the lawn of its isolation host *Aeromonas salmonicida* JK, phage JELG-KS1 demonstrated a range of possible plaque sizes and appearances in the double-agar overlay assays with 0.7% LB as the top medium (Figure 1). Phage JELG-KS1 plaque appearance as well as its possible plaque diameter range was found to be dependent on the double-agar overlay assay plate incubation temperature, although the efficiency of plating (observed plaque count) did not differ between double-agar overlay assays incubated overnight at RT, 30 °C, and 37 °C (Kruskal–Wallis test; *p* = 0.34). After incubation at RT and 30 °C, the plaques were clear and large, whereas on a plate incubated at 37 °C, the plaques were small, with some of them demonstrating remarkable turbidity.

Interestingly, at 30 °C, plaques could even reach a diameter of ~4 mm, but pinpoint phage-resistant host colonies formed within the plaques, unlike at the other incubation temperatures. Most of the JELG-KS1 plaques demonstrated translucent haloes at the periphery of the plaque regardless of the incubation temperature, which might be suggestive of polysaccharide depolymerase activity associated with the virion components. Phage JELG-KS1 could not form plaques in the double-agar overlay assay incubated at 4 °C, although the bacteria were able to grow very slowly and eventually formed a lawn at this temperature. Incubation of double-agar overlay plates at 37 °C was chosen for routine experimentation with phage JELG-KS1 on the lawn of its isolation host, as the consistently smaller plaque sizes allowed for easier plaque enumeration, despite the downside of plaque turbidity, which was especially relevant for the spot-test experiments.

*Aeromonas* phage JELG-KS1 virions demonstrate a typical podophage morphology—icosahedral capsid with a short tail. Morphological feature measurements across multiple randomly chosen intact virions from different fields of view allowed us to determine the capsid diameter of JELG-KS1 as 62.1 ± 3.5 nm and the tail length as 14.7 ± 2.1 nm (Figure 1D).

### 3.2. Infective Properties and Stability of the JELG-KS1 Phage

Adsorption assays revealed that, in our experimental setting, more than half of the JELG-KS1 virions adsorbed onto the surface of the host cells within the first minute, with less than 5% of unadsorbed virions remaining in the sample five minutes after the infection of *A. salmonicida* JK culture by phage JELG-KS1 (Figure 2A).

Repeated one-step growth experiments with phage JELG-KS1 and *A. salmonicida* JK revealed that the JELG-KS1 phage has a latent period of 24 min while the rise phase takes 24 more minutes, and, on average, each infected cell releases 71 ± 12 progeny virions upon bursting (Figure 2B).

Growth curves of *A. salmonicida* JK with the addition of phage JELG-KS1 at different multiplicities of infection showed that phage JELG-KS1 is capable of efficiently killing the host population, even at lower MOIs. For example, clearance of the host cell culture was observed at 3.5 h post-infection with a phage-to-host cell ratio as low as 1 to 5000 (Figure 2C). The lack of increase in optical density after phage treatment might suggest that no phage-resistant JK host strain mutants emerged and started to grow during the experiment, which could mean that resistance to JELG-KS1 is costly for the host. However, to check the frequency of JELG-KS1 resistance occurrence in strain JK cells, the experiment should have been conducted over a much longer period. 

Incubation of phage JELG-KS1 at different temperatures for an hour revealed that there was no significant difference between the PFU/mL count for samples incubated either at RT or 35 °C. However, incubation at 40 °C revealed a drop in the remaining PFU/mL count, with further incubation temperature increases of +5 °C increments making differences with the sample incubated at RT more and more pronounced, reaching a complete loss of PFUs after incubation of the phage sample at 55 °C degrees for an hour (Figure 2D).

As to the stability of JELG-KS1 virions in solutions of different pHs, no significant reduction in the PFU/mL count was revealed in the pH range of 5–10 (compared with pH 7, which served as the control). Exposure of the JELG-KS1 sample to acidic pH 4 for an hour resulted in a more than 10,000-fold decrease in infective virions, while exposure to basic pH 11 had a less detrimental effect. A complete reduction in the infectious phage titer was observed after an hour-long exposure to pH 3 and pH 12 (Figure 2E).

The ability of JELG-KS1 to quickly adsorb to the host cell surface in addition to a relatively short latent period of 24 min (after which the host cells start to burst, releasing around ~70 progeny phages per lysed cell), coupled with its stability, even in a range of temperatures and environmental pH conditions exceeding those the phage might encounter in an aquaculture setting, would have made JELG-KS1 a worthwhile candidate for testing its capabilities for the biocontrol of *Aeromonas* spp. in local aquaculture. This, however, seems to be precluded by a host-range screen that revealed the failure of JELG-KS1 to lyse any other *Aeromonas* spp. strain it was tested against except for its isolation host, *A. salmonicida* JK (Figure 3). The ability of the host strain of phage JELG-KS1 to grow at 37 °C, coupled with the 16S rRNA and *gyrB* gene phylogenies (Figure 3), allows to assume that the JK strain is a mesophilic strain of *A. salmonicida*. The main drawback of our host-range screen is that, despite phage JELG-KS1 being tested against 20 *Aeromonas* spp. strains representing seven different species, only 7 of the strains (6, if the isolation host is excluded) represented *A. salmonicida*. While JELG-KS1 does not seem to be able to infect across species of the genus *Aeromonas*, testing it against a larger number of *A. salmonicida*—specifically, atypical mesophilic strains—in the future might show how narrow its host range actually is within the more realistic boundary of a single species.

### 3.3. Complete Genome of Aeromonas Phage JELG-KS1

The complete genome of *Aeromonas* podophage JELG-KS1 was determined to be a 40,505 bp long dsDNA molecule with 185 bp short direct terminal repeats. The genome of JELG-KS1 was predicted to have 53 open reading frames (ORFs), all located on a direct strand, resulting in a coding capacity of 92.14%. ATG is thought to serve as a start codon for 47 of the ORFs, TTG for 3 of them, GTG for 2, and CTG was predicted as the start codon for a single ORF. No tRNAs were predicted in the genome of JELG-KS1. Products of only 23 out of the 53 ORFs remained without a function prediction after extensive annotation using comparative genomics approaches (Appendix A). The podoviral morphology, coupled with a single gene coding for a DNA-dependent RNA polymerase, immediately gave a hint that JELG-KS1 is very likely a representative of the *Autographiviridae* phage family.

### 3.4. Intergenomic Similarity to Other Phages

A BLASTn search revealed that the complete genome nucleotide sequence of JELG-KS1 is rather unique as of today; intergenomic similarity to the most closely related phages found in GenBank did not exceed ~40%. Under the demarcation criteria currently adopted by the ICTV’s Bacterial Virus Subcommittee, JELG-KS1 represents not only a novel phage species but also might serve as an exemplar isolate of a novel phage genus, which is yet to be proposed (Figure 4). 

The highest-scoring hits were all to either taxonomically recognized or tentative representatives of the *Autographiviridae* family, *Studiervirinae* subfamily. The closest hits officially classified at the genus level were to phages from the *Chatterjeevirus* and *Ningirsuvirus* phage genera. Bacteriophage JELG-KS1 was, unsurprisingly, most similar (~40% intergenomic similarity) to several of the *Aeromonas* phages, all of which are still taxonomically unrecognized at the genus level but tentatively represent the same genus within the *Studiervirinae* subfamily (*Aeromonas* phages T7-Ah [82], PZL-Ah152 [83], vB_AhaP_PT2 (not described in a dedicated article to the best of our knowledge), and PZL-Ah8 [84]). 

These hits to several of the *Aeromonas* phages were followed by hits to three *Escherichia* phages (vB_Eco_Mak, vB_Eco_Bam, and vB_Eco_Titus; intergenomic similarity of ~32%) that not only represent a tentative genus but also might be considered isolates of the same phage species, and lots of *Vibrio*-infecting phages from the genus *Chatterjeevirus* (intergenomic similarity of ~30%). To a lesser extent, the JELG-KS1 complete genome nucleotide sequence is also reminiscent of representatives from the genus *Ningirsuvirus* (*Dickeya* phages Ninurta, DchS19, vB_DsoP_JA10, *Klebsiella* phages vB_KpnP_Sibilus, vB_KPnP_NahiliMali, and *Erwinia* phage pEp_SNUABM_12), with whom it shows an intergenomic similarity of ~21%.

### 3.5. Selected Marker Protein Phylogeny Reconstruction

DNA-dependent RNA polymerase, the major capsid protein, DNA polymerase, and the terminase large-subunit amino acid sequences of the JELG-KS1 phage were chosen as marker proteins to reconstruct individual protein phylogenies with the most similar homologous proteins of other phages (Figure 5).

In all four of the ML trees, respective proteins of JELG-KS1 reliably and monophyletically clustered together with homologs from the previously identified most intergenomically similar phages. The clade into which sequences of JELG-KS1 marker proteins fell comprised *Vibrio*-infecting phages from the *Chatterjeevirus* genus, previously mentioned *Aeromonas* phages representing a tentative novel phage genus, and the *Escherichia* phages Mak, Bam, and Titus. Unsurprisingly, given their intergenomic distances, which indicated that these *Escherichia* phages might be isolates of the same species, their amino acid sequences showed little to no pairwise differences between them. The sequences of JELG-KS1 proteins, nevertheless, were at the tips of rather long external branches in each case, indicating substantial divergence, even from the closest homologs yet uncovered. It was also noted that the context of other sequences most closely related to those of JELG-KS1 outside this large clade differed across the trees.

### 3.6. Proteome-Based Clustering with Other Phages

Proteome-based clustering using vConTACT2 under default settings using 18,554 phage genomes found in the INPHARED October 2022 release revealed that there are 39 first neighbors of *Aeromonas* phage JELG-KS1 (Appendix A). These neighbors were exclusively *Autographiviridae* podophages and represented two viral clusters (arbitrarily designated as VC_83_0 and VC_84_0); additionally, several phages (*Delftia* phage IME-DE1 (KR153873), *Ralstonia* phage RPSC1 (MF893341) and *Vibrio* phage Rostov 13 (partial genome split into OK169294 and OK169295)) represented outliers among the first neighbors of JELG-KS1. Consistent with the results of previous analyses, JELG-KS1 clustered into the VC_83_0 together with *Vibrio*-infecting *Chatterjeevirus* representatives (except for the *Vibrio* phage Rostov 13, which might be due to its publicly available genome being split into two accessions for some reason), and the same *Aeromonas* and *Escherichia* phages tentatively representing novel phage genera (Figure 6). In contrast, cluster VC_84_0 comprised *Ralstonia* phages philTL-1, RpT1, RpY2, RSB2. This further removed any doubt that *Aeromonas* phage JELG-KS1 might be considered a representative of the *Studiervirinae* phage subfamily.

### 3.7. Comparison with Representatives of the Most Closely-Related Phage Genera

Based on the pairwise complete genome nucleotide sequence intergenomic distances, exemplar isolates of the selected phage species within either recognized genera (*Chatterjeevirus ICP3*, represented by *Vibrio* phage ICP3 (HQ641340; [85]), *Ningirsuvirus Ninurta*, represented by *Dickeya* phage Ninurta (NC_047964; [86])) or tentative phage genera (*Aeromonas* phage T7-Ah (MT740748; [82]), *Escherichia* phage vB_Eco_Titus (OX090892; not yet published to the best of our knowledge)) that were evolutionarily related to *Aeromonas* phage JELG-KS1 were selected for a comparative genomics context.

Roary analysis under the rather liberal shared gene product identification criterion of 30% protein similarity has resulted in the identification of up to 22 products common to all five of the selected phages. Nearly all the functionally annotated JELG-KS1 structural or morphogenesis proteins were also found in other phages at the 30% similarity criterion (except for the putative tail-assembly protein encoded by ORF33, and the putative decoration protein encoded by ORF49). Similarly, of the proteins categorized as being involved in DNA recombination, modification, and repair, only the product of ORF9, encoding a DNA ligase, was not found to be shared by all five of the selected phages at this similarity threshold. Additionally, products of ORF18 (ssDNA-binding protein), ORF20 (endolysin), ORF25 (hypothetical protein), ORF30 (DUF2717-containing protein), and ORF45 (holin) were found to be shared by the phages analyzed. Despite Roary authors advising not to use low BLASTp percentage identities, as anything might cluster with anything under lower identity threshold criteria, and stating that Roary is not intended for comparing extremely diverse sets of genomes, we find that using the tool this way is helpful for gaining quick insights about what to look at in detail further on (Appendix A). At the 50% BLASTp identity set for Roary, the identified shared-product count dropped to 13, whereas at 70% identity, only the terminase large subunit encoded by ORF51 was found to be “shared” between these phages.

Clinker was further used to perform a more precise and informative selected phage protein comparison that could be visualized with a more pronounced focus on JELG-KS1 (Figure 7). As expected, global alignments performed by clinker before calculating the identities resulted in some of the proteins being determined by Roary as shared between the phages (e.g., several virion morphogenesis module products) falling under the minimal 30% identity threshold in clinker analysis.

*Aeromonas* phage JELG-KS1 has demonstrated a conserved modular genome organization typically seen in other subfamily *Studiervirinae* podophages (e.g., *Chatterjeevirus ICP3* [85], *Ningirsuvirus ninurta* [86], *Pektosvirus PP81* [87,88], *Teseptimavirus T7* [89], *Ghunavirus gh1* [90]) and tentative *Studiervirinae* phages such as *Pseudomonas* phage Eir4 [91], *Hafnia* phage Ca [92], etc. An open reading frame encoding a DNA-directed RNA polymerase, a hallmark of *Autographiviridae* podophages, was readily identifiable among the first few ORFs at the beginning of the genome, which likely represent early genes. Next, a gene module comprising largely genes involved in DNA recombination, modification, and repair (but having an endolysin-coding ORF within it) was identifiable. Roughly, the second half of the genome hosted ORFs encoding proteins that are mainly involved in JELG-KS1 virion morphogenesis. The remaining genes encoding proteins involved in Gram-negative host lysis at the end of the JELG-KS1 lytic cycle (holin and spanins) were located near the end of the JELG-KS1 genome, in the proximity of the tail fiber and both terminase-subunit-encoding ORFs, more than 24 kbp away from the endolysin-encoding ORF (Figure 7). 

Expectedly, most of the *Aeromonas* phage JELG-KS1 products with a functional assignment had homologs in related phages and showed greater similarity to their counterparts from *Aeromonas* phage T7-Ah. The genome of JELG-KS1 also differed from its relatives in the presence or absence of some of the hypothetical-protein-encoding ORFs; most of these differences were confined to the DNA recombination, modification, and repair gene module. The only gene order rearrangement relative to the most closely related phages was noted in the presumed “early gene” module. Namely, a protein that was identified as a host cell RNA polymerase inhibitor in JELG-KS1 (ORF10 product, UTQ78146.1) was found between DNA ligase- and deoxynucleoside monophosphate kinase-encoding ORFs (ORF9 and ORF11, respectively), whereas in *Aeromonas* phage T7-Ah and *Escherichia* phage vB_Eco_Titus, an unannotated protein showing amino acid similarity to the JELG-KS1 ORF10 product was located after the presumed JELG-KS1 ORF11 analog. 

Easyfig visualization of the genomic nucleotide sequence similarities further showed that despite the detectable protein amino acid similarities, in many cases, these are not evident from the underlying nucleotide stretches alone. However, the addition of translated genomic nucleotide sequence comparisons provided an even finer resolution that complements a simple protein aa vs. an aa sequence comparison (e.g., whether some of the ORFs were just not called because of the differences in the gene prediction algorithms used, and what regions of the proteins might be similar to each other in the case of the full-length global similarity threshold not being met between them; Figure 7 and Appendix A).

This allowed us to note that, despite the lack of global product aa sequence similarity as well as varying product lengths (579 aa in ICP3 to 837 aa in JELG-KS1), tail-fiber proteins of these phages show conservation of the first ~180 N-terminal amino acids corresponding to a conserved phage T7 tail-fiber domain (pfam03906). From this, it seems reasonable to assume that JELG-KS1 might have a different adsorption receptor or suitable receptor repertoire than that of its most closely related phages. 

Additionally, plasticity in gene contents was noticed for the “early gene” region at the beginning of the genomes of these phages, which shows a lack of meaningful conservation. In this way, a nucleotide and translated nucleotide sequence comparison allows to carefully speculate that *Studiervirinae* subfamily phages might be rather promiscuous regarding what is encoded by the genome part before the DNA-dependent RNA-polymerase-encoding ORF, and that these differences cannot be attributed to the diverse ORF calling algorithms/curation criteria used by the phage genome annotation authors.

## 4. Conclusions

The isolation, characterization, and complete genome elucidation of the novel *Aeromonas salmonicida*-infecting podophage JELG-KS1 presented in this study expands the current knowledge on the diversity of phages infecting bacteria belonging to the genus *Aeromonas*; more specifically, mesophilic *Aeromonas salmonicida*.

The novel *Aeromonas* phage JELG-KS1 demonstrated multiple properties (strictly lytic, not encoding for any known virulence genes, short latent period, moderate burst size, and virion stability under unfavorable environmental conditions) that are highly desired in candidate phages to be used for the biocontrol of *Aeromonas* spp. in an aquaculture setting. The feasibility of its potential practical applications, however, is immediately hampered by the narrowness of its host range, which was confined solely to its isolation host, *A. salmonicida* JK, previously retrieved from wastewater. 

Strain-specificity and the inability of phage JELG-KS1 to lyse any of the aquaculture-associated *Aeromonas* spp. strains it was tested against, including several *Aeromonas salmonicida* strains isolated from diseased fish, further cautions that the isolation of novel bacteriophages for a potential practical application in biocontrol should better be carried out using relevant host strains associated with the setting in which the biocontrol application is to eventually take place to reduce the risk of isolating a priori highly host-specific phages associated with irrelevant host strains. 

Thus, regardless of the properties a particular phage demonstrates on its isolation host, no phage should be proposed as a potential biocontrol or therapeutical agent before it has successfully passed a productive infection challenge against a panel comprising multiple relevant target strains.

## Figures and Tables

**Figure 1 microorganisms-12-00542-f001:**
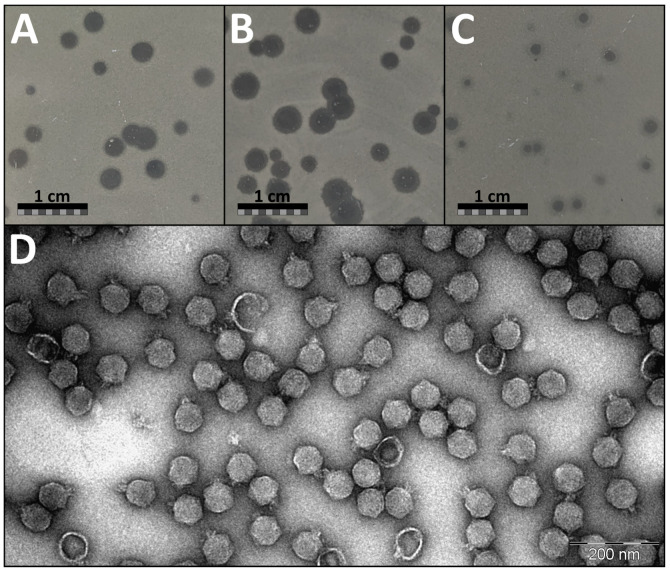
Plaque and virion morphology of *Aeromonas* phage JELG-KS1. Top row: Photographs of the representative phage JELG-KS1 plaques after incubating the double-agar overlay assay with *Aeromonas salmonicida* JK culture as the lawn for 24 h at (**A**) RT, (**B**) 30 °C, and (**C**) 37 °C (bottom-layer agar—1.5% Peptone Mix; top-layer agar—0.7% LB); scale bars represent 1 cm. (**D**) Transmission electron micrograph showing a representative field of view for a CsCl-purified *Aeromonas* phage JELG-KS1 sample negatively stained with 0.5% uranyl acetate; the scale bar represents 200 nm.

**Figure 2 microorganisms-12-00542-f002:**
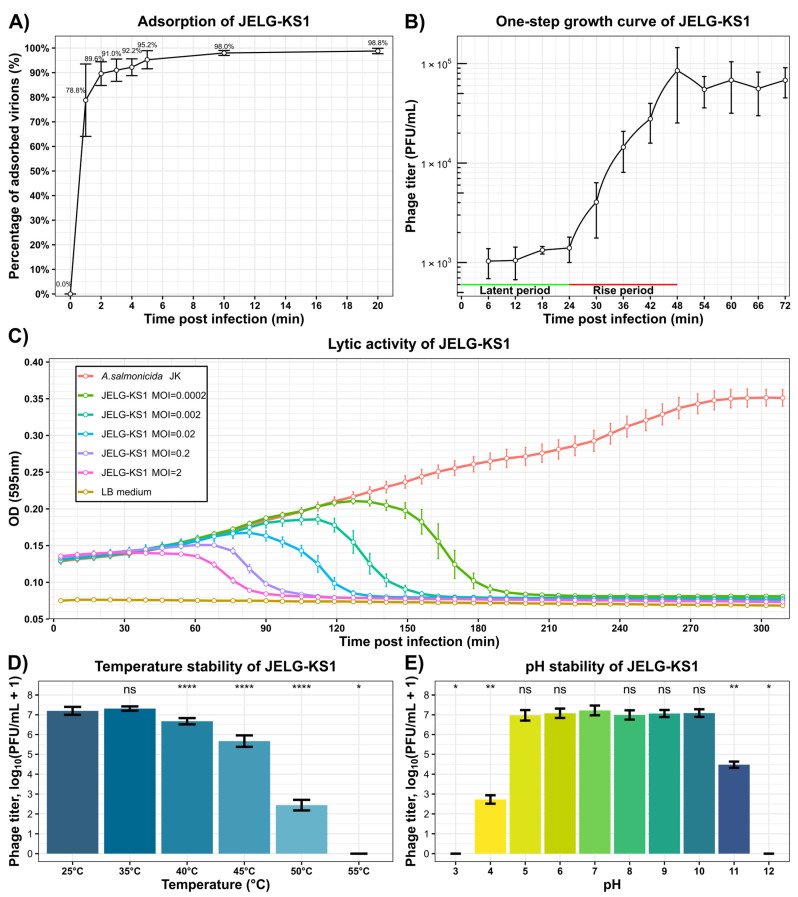
Infective properties and stability of *Aeromonas* phage JELG-KS1. (**A**) Adsorption curve of phage JELG-KS1 virions to the host cells of *A. salmonicida* strain JK at RT. (**B**) One-step growth curve of phage JELG-KS1 using *A. salmonicida* strain JK as a host. The green horizontal line indicates the latent period; the red horizontal line indicates the rise period. (**C**) Effects of phage JELG-KS1 lytic activity on the *A. salmonicida* JK cell growth at RT. Curve color corresponds to the condition being tested according to the legend (red—*A. salmonicida* JK cells without addition of phage; orange—plain LB medium; other curves—*A. salmonicida* JK + phage JELG-KS1 at different multiplicities of infection). (**D**) Temperature and (**E**) pH stability of the phage JELG-KS1 virions after the respective treatment for an hour. Statistical significance of PFU/mL differences in comparison to the controls (25 °C and pH 7, respectively) is indicated above the bars (Wilcoxon rank–sum test with Benjamini–Hochberg *p*-value adjustment for multiple comparisons; ns—not significant, *—adj. *p* < 0.05, **—adj. *p* < 0.01, ****—adj. *p* < 0.0001). In all the tiles, except for the one-step growth curve, the results shown represent the mean of at least three independent replicates ± one standard deviation. For the one-step growth curve, a representative experiment is shown, and values represent the mean of at least three technical replicates ± one standard deviation.

**Figure 3 microorganisms-12-00542-f003:**
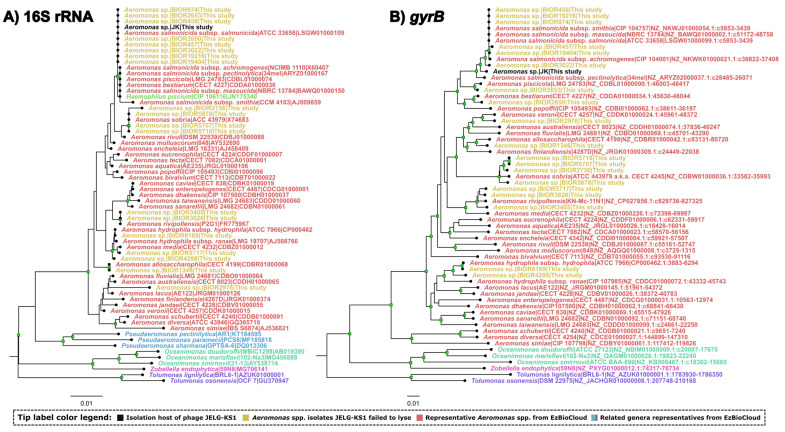
16S rRNA (**A**) and *gyrB* (**B**) gene sequence neighbor-joining tree validating the identity of the strains that phage JELG-KS1 was tested against. The analyses involved 65 (**A**) and 61 (**B**) nucleotide sequences (near-complete 16S rRNA and *gyrB* gene sequences from bacterial isolate JK (isolation host of phage JELG-KS1), 19 *Aeromonas* spp. from the BIOR collection that phage JELG-KS1 was tested against and failed to lyse, and either 45 (**A**) or 41 (**B**) other closely-related bacterial species (for context). In both of the trees, labels of closely related bacterial species correspond to the taxa and are in the format of “Species|Strain|Accession” (with the addition of nucleotide coordinates of the respective accession for *gyrB* sequences after the colon; “c” before the coordinates indicates the complementary strand); tip-label colors correspond to different bacterial genera or sequence groups of interest according to the legend. Branches having bootstrap support higher than or equal to 80% have their distal nodes indicated by green rectangles. The evolutionary distances are in the number of base differences per site, and the tree is drawn to scale. Sequences were aligned using MAFFT (L-INS-i mode) and the MSAs were trimmed using gblocks under the default parameters. The trimmed input alignments used for the NJ tree generation had 1213 columns, 192 distinct patterns, 153 parsimony-informative, 55 singleton sites, and 1005 constant sites for the 16S rRNA MSA and 2166 columns, 887 distinct patterns, 918 parsimony-informative, 112 singleton sites, and 1136 constant sites for the *gyrB* MSA, respectively. The trees were rooted at the midpoint and visualized in FigTree.

**Figure 4 microorganisms-12-00542-f004:**
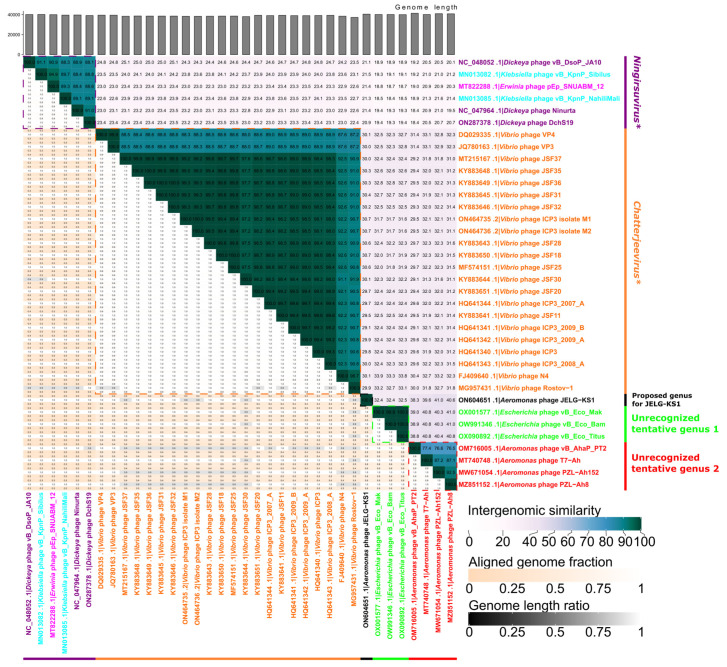
Intergenomic similarities between *Aeromonas* phage JELG-KS1 and the most-similar phages completely sequenced to date. An annotated VIRIDIC heatmap showing pairwise intergenomic similarities (%) of *Aeromonas* phage JELG-KS1 and other sequenced phages related to it (n = 36) is depicted. Labels of phages infecting different host genera are colored differently arbitrarily, but the label of JELG-KS1 is in black. The heatmap is annotated at the pairwise intergenomic similarity level corresponding to the demarcation of phage genera (≥70%) using dashed squares and bars next to the corresponding labels; bars are labeled based on the presence of isolates representing ICTV-recognized phage species within the genus-level clusters if applicable (denoted by an asterisk after the respective italicized label).

**Figure 5 microorganisms-12-00542-f005:**
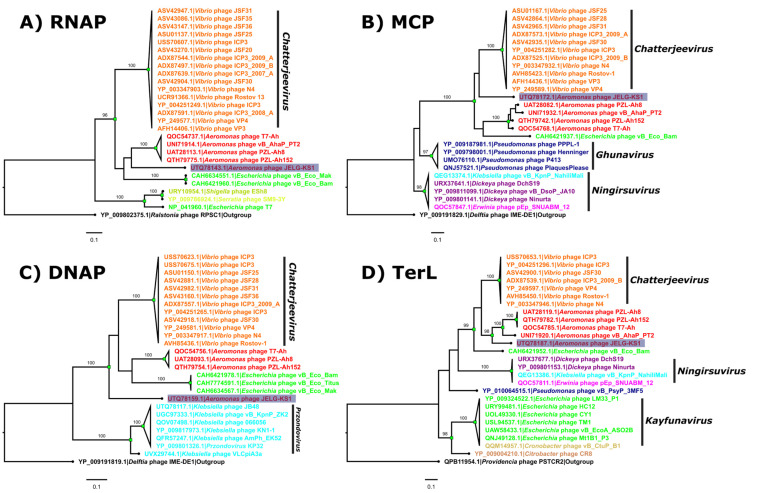
Maximum-likelihood trees of the selected *Aeromonas* JELG-KS1 phage marker protein amino acid sequences. Maximum-likelihood trees of the selected *Aeromonas* JELG-KS1 phage protein amino acid sequences and the 25 most related non-redundant sequences originating from the proteomes of other completely sequenced phages are shown: (**A**) DNA-dependent RNA polymerase (RNAP); (**B**) major capsid protein (MCP); (**C**) DNA polymerase (DNAP); (**D**) terminase large subunit (TerL). All the trees are drawn to their respective scales, and branch lengths correspond to the number of amino acid differences per site. Tips are labeled as “protein accession|originating phage” and are colored based on the given phage host genus. In all the trees, the tip containing the respective protein sequence of JELG-KS1 is highlighted in blue. The trees are outgroup-rooted (additional outgroup identifier in the corresponding tip labels; outgroup tip labels are colored in black). The distal nodes of branches with ≥95% ultrafast bootstrap (UFBoot) support (out of 1000 replicates) are indicated by green squares, and the percentage of UFBoot support is indicated for such branches. Well-supported clades containing sequences of ≥3 proteins from different phages, including at least one phage that is taxonomically recognized at the genus level, are annotated accordingly, extending the genus annotation to all the sequences originating from the same shared MRCA node. Descriptions of the selected phage marker protein amino acid sequence datasets and the generated MSAs, as well as features of the trees built, can be found in Appendix A.

**Figure 6 microorganisms-12-00542-f006:**
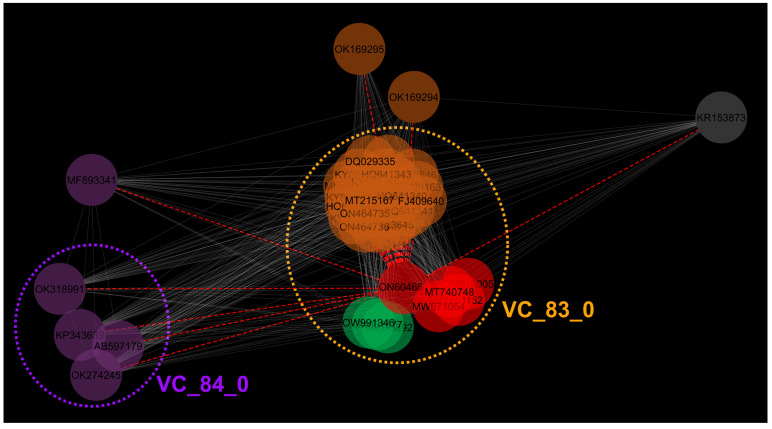
Proteome-based clustering of *Aeromonas* phage JELG-KS1 (ON604651) with other completely sequenced phages. Sub-network containing vConTACT2-identified first neighbors of the *Aeromonas* phage JELG-KS1 visualized under edge-weighted spring-embedded layout is shown (n = 40). Phages are represented by nodes that are colored according to their indicated hosts as follows: red—*Aeromonas* spp.; green—*Escherichia* spp.; brown—*Vibrio* spp.; purple—*Ralstonia* spp.; grey—*Delftia* spp. Phages belonging to vConTACT2-identified clusters are enclosed by dotted ellipses. Dashed red edges represent links to the *Aeromonas* phage JELG-KS1. See Appendix A for additional details.

**Figure 7 microorganisms-12-00542-f007:**
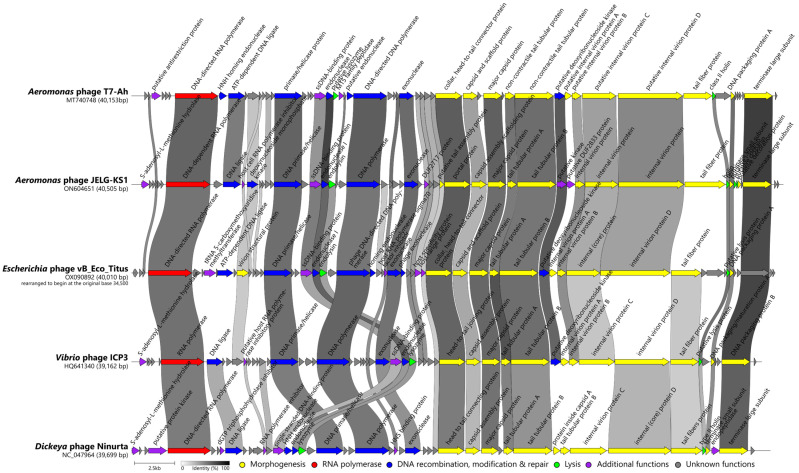
Genome organization and proteome content comparison of *Aeromonas* phage JELG-KS1 and selected related phages. Genomes are drawn to scale; the scale bar indicates 2500 base pairs. Arrows representing open reading frames point in the direction of the transcription and are color-coded based on the function of their putative product according to the legend. Slanted labels above the arrows indicate the predicted function for the given ORF putative product in the case where it had a function assigned (original annotations from downloaded GenBank files were retained). Ribbons connect phage proteins sharing >30% amino acid sequence similarity and are colored in a gradient from white to black according to their percentage identity.

**Table 1 microorganisms-12-00542-t001:** Details of the additional bacterial strains used for *Aeromonas* phage JELG-KS1 host range screening.

Strain Accession No. (BIOR)	Isolation Date	Strain Isolation Source	Species
1346-2016-VM	19 October 2016	Sturgeon (*Acipenser sturio*)	*Aeromonas allosaccharophila*
2650-2017-VM	19 April 2017	Salmon (*Salmo salar*)	*Aeromonas bestiarum*
4288-2017-VM	24 November 2017	Carp (*Cyprinus carpio*)	*Aeromonas hydrophila*
6169-2018-VM	16 July 2018	Rainbow Trout (*Oncorhynchus mykiss*)	*Aeromonas hydrophila*
2653-2017-VM	19 April 2017	Brown Trout (*Salmo trutta*)	*Aeromonas piscicola*
3405-2017-VM	7 August 2017	Salmon (*Salmo salar*)	*Aeromonas rivipollensis*
3626-2017-VM	21 August 2017	Brown Trout (*Salmo trutta*)	*Aeromonas rivipollensis*
5717-2018-VM	16 May 2018	Whitefish (*Coregonus lavaretus*)	*Aeromonas rivipollensis*
457-2016-VM	8 April 2016	Brown Trout (*Salmo trutta*)	*Aeromonas salmonicida*
458-2016-VM	8 April 2016	Brown Trout (*Salmo trutta*)	*Aeromonas salmonicida*
974-2016-VM	5 August 2016	Rainbow Trout (*Oncorhynchus mykiss*)	*Aeromonas salmonicida*
3022-2017-VM	19 June 2017	Salmon (*Salmo salar*)	*Aeromonas salmonicida*
19216-2022-VM	23 September 2022	Brown Trout (*Salmo trutta*)	*Aeromonas salmonicida*
19404-2022-VM	14 October 2022	Salmon (*Salmo salar*)	*Aeromonas salmonicida*
2736-2017-VM	8 May 2017	Tench (*Tinca tinca*)	*Aeromonas sobria*
5718-2018-VM	21 May 2018	Salmon (*Salmo salar*)	*Aeromonas sobria*
5707-2018-VM	25 May 2018	Salmon (*Salmo salar*)	*Aeromonas sobria*
5878-2018-VM	1 June 2018	Whitefish (*Coregonus lavaretus*)	*Aeromonas sobria*
2976-2017-VM	16 July 2017	Brown Trout (*Salmo trutta*)	*Aeromonas veronii*

**Table 2 microorganisms-12-00542-t002:** Primers used for the *gyrB* gene sequencing of the *Aeromonas* spp. used in this study. The third column indicates nucleotide coordinates within the *A. salmonicida* subsp. *salmonicida*-type strain ATCC 33658 *gyrB* gene (LSGW01000099.1:c5853-3439).

Primer Name	Primer Sequence (5′-3′)	Nucleotide Coordinates within the *gyrB* Gene
Fw1_Aeromonas_gyrB	AAGCGCCCGGGGATGTA	64–80
Rv1_Aeromonas_gyrB	GTCCGGGTTGTACTCGTC	c1461-1444
Fw2_Aeromonas_gyrB	GCCCGTTTCGACAAGATGATC	1369–1389
Rv2_Aeromonas_gyrB	CGCAGGGCGTTGGTCTC	c2393-2377

## Data Availability

The complete annotated genome of *Aeromonas* phage JELG-KS1 has been deposited into GenBank and is publicly available under accession number ON604651. All other relevant data and accession numbers of the sequences used for the analyses are found within the paper and its Appendix A.

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
