# Peer review of "Isolation and Characterization of a Novel Aeromonas salmonicida-Infecting Studiervirinae Bacteriophage, JELG-KS1"

_microorganisms, 2024, doi:10.3390/microorganisms12030542_

Round 1
Reviewer 1 Report
Comments and Suggestions for Authors
Comments and Suggestions for Authors
In the provided manuscript, “Isolation and characterization of novel Aeromonas salmonicida infecting Studiervirinae bacteriophage JELG-KS1”, authors isolated, identified and characterized a new phage JELG-KS1 specific to Aeromonas salmonicida. In general, the manuscript is well organized. The methods are described in great detail and the conclusions are supported by the results obtained. At the same time, some improvements need to be made, mainly concerning Aeromonas host strain isolation and taxonomy identification.
Main suggestions:
Introduction, Lines 88-89. The data obtained from the Genbank should be updated. New Aeromonas phages were probably discovered last two years (2022-2023). In addition, the dates of access to various software and databases in the text of the manuscript should be refreshed whenever possible.
Section 2.8 contains two different points, one of them is the definition of the host range, and the second is the taxonomic identification of the host strain. Section 2.8 should be divided into two sections. Also, some data are results, but not methods. It was not possible to find an appropriate section in the results, including the isolation and characterization of the Aeromonas host strain. Please add this section and relocate the data from methods.
Please, explain why gyrB gene was chosen as an additional gene from Aeromonas strain JK taxonomy identification, has it good discriminating ability for the genus Aeromonas?
Another unclear point is how other Aeromonas strains were taxonomically identified (Table 1)? Was it 16SpPHK sequencing or some other method? Please, clarify this.
Technical comments:
Abstract, Line 15. “genomic analysis indicates” not “indicate”.
Section 2.9. Lines 340-343 are reduntant. May be shortened.
Section 3.1. Lenes 448-450. Please, clarify or re-write the phrase “sufficiently broad range of possible plaque morphologies”.
Materials and Methods, Line 170. Please, add the manufacturer of the orbital shaker.
Author Response
Reviewer 1
Comments and Suggestions for Authors
In the provided manuscript, “Isolation and characterization of novel Aeromonas salmonicida infecting Studiervirinae bacteriophage JELG-KS1”, authors isolated, identified, and characterized a new phage JELG-KS1 specific to Aeromonas salmonicida. In general, the manuscript is well organized. The methods are described in great detail and the conclusions are supported by the results obtained. At the same time, some improvements need to be made, mainly concerning Aeromonas host strain isolation and taxonomy identification.
Author response: We would like to thank Reviewer 1 for reading through the manuscript and pointing out several ways how we could improve the manuscript further! Please, find our point-by-point response below. We have additionally tried to read through the manuscript once more to fix small writing mistakes and typos we could spot. Should any of the points remain unaddressed adequately after this round of revisions – we are happy to address the remaining issues during the second round of revisions, if necessary!
Reviewer 1: Introduction, Lines 88-89. The data obtained from the Genbank should be updated. New Aeromonas phages were probably discovered last two years (2022-2023). In addition, the dates of access to various software and databases in the text of the manuscript should be refreshed whenever possible.
Author response: We thank Reviewer 1 for this very reasonable suggestion. This manuscript has indeed been in preparation for a rather long time and the information provided in the initial version of Supplementary Table 1, indeed, could be viewed as very out of date given how quickly the field of bacteriophage discovery progresses. The table has now been updated according to the Aeromonas phage genomes found in GenBank on the 8th of February, 2024 (also updating information in one of the columns to include the latest currently available ICTV Virus Metadata Resource which has also been updated). This now ensures that the information provided within the relevant paragraph of the introduction (Lines 87-95) and the table is up to date of the submission of this manuscript. Whereas the initial version of the table listed 139 phages, the revised version now includes information about 178 phages for which Aeromonas spp. were indicated as hosts based on the complete genome-associated metadata.
The dates of access to various software and databases in the text of the manuscript corresponds to either the date when the particular software was installed on the workstation the relevant analysis was performed on, or when a particular database was accessed to perform the task outlined. Updating to the different, more recent versions of the databases would necessitate performing a lot of the analyses outlined from scratch. Before finalization and submission of this manuscript we have checked (in late January, 2024) whether there is a need to update such analyses, but found that no additional objects directly relevant to the results presented herein became available at the GenBank since we performed our comparative genomics, etc. Thus, reanalysis would not actually impact the results presented in any meaningful way, although with the pace of advancement of the phage diversity research, as more phages get described, phages more similar to JELG-KS1 might indeed eventually get isolated and sequenced.
Reviewer 1: Section 2.8 contains two different points, one of them is the definition of the host range, and the second is the taxonomic identification of the host strain. Section 2.8 should be divided into two sections. Also, some data are results, but not methods. It was not possible to find an appropriate section in the results, including the isolation and characterization of the Aeromonas host strain. Please add this section and relocate the data from methods.
Author response: This section indeed contains two different points (the actual experimentation on whether JELG-KS1 gives productive infection on the Aeromonas spp. strains, but also the verification of the procured Aeromonas spp. strain identity using molecular markers, such as 16S rRNA and gyrB, sequencing.), so to say, but was written in such a way intentionally.
To elaborate, given the timespan of their isolation, these “historic” isolates were initially characterized as Aeromonas spp. using different commercial biochemical tests (e.g., API 20 NE) and later on MALDI-TOF, and stored in the respective local strain collection without any particular interest in them for a long time, thus, there was little comparability between the approaches of their identification, in addition to most of them performing sub-optimally and not allowing to discriminate between the closely related Aeromonas species. Although the precise tests and versions of the software relevant to the initial identification of each strain, as well as the results of such identifications can probably be found, we believe that this would unnecessarily overload this part of the work without adding up much to it.
Nevertheless, although there was no doubt that the strains acquired are all Aeromonas spp. and JELG-KS1 failed to lyse any of them, we still thought that it would be better to “be safe than sorry” and went for a more unified and universal approach. Thus, within this study, we have opted to proceed with a more precise identification of these strains utilizing Sanger-based sequencing (at first traditional 16S, then gyrB gene sequence, were chosen as markers). Because of this, the section 2.8. was written to contain both these methodology topics as they represent the actual workflow that has happened experimentally and preserve the actual “flow” of the study.
We would like to preserve section 2.8. as it is seen in the initial version of the manuscript as we do not see how to adequately rework it while preserving the original “course of events”, however, we will comply with the request of dividing it into two separate methodology points if our reasoning is deemed to be inappropriate.
Results regarding the identity of the JELG-KS1 isolation host strain are provided within lines 533-535 (“The ability of the host strain of the phage JELG-KS1 to grow at 37 °C coupled with the 16S rRNA and gyrB gene phylogenies (Fig 3) allows to assume that the strain JK is a mesophilic strain of A. salmonicida.”) and in a visual manner in Figure 3 which is located nearby and is referenced therein.
We are not sure what particularly was meant regarding the data representing results and not methods. If this was meant to be a commentary about the Table 1 contents being in this section – we view these other strains as “materials” rather than results. We do not oppose reworking this and moving parts perceived as results from methodology into the result and discussion section, but after internal discussion between the co-authors – we could not unambiguously reach a consensus on what actual parts is this applicable to. If the respected Reviewer 1 could elaborate on this point more specifically, we will try to comply in the second round of revisions.
Reviewer 1: Please, explain why gyrB gene was chosen as an additional gene from Aeromonas strain JK taxonomy identification, has it good discriminating ability for the genus Aeromonas?
Author response: Many previous studies have concluded that identification of Aeromonas species is quite challenging due to genetic and biochemical heterogenicity (even among the copies of rRNA operons of the same bacteria) and insufficient interspecies variation in the widely used 16S rRNA gene sequences (Küpfer et al., 2006), (Persson et al., 2015), (Fernández-Bravo & Figueras, 2020). It has been shown that sequences of several housekeeping genes such as GyrB used in the study provide more nuance and ability to differentiate tight taxonomic groups, especially when characterizing Aeromonas species (Navarro & Martínez-Murcia, 2018). Aeromonas media complex (Talagrand-Reboul et al., 2017) highlights the importance of using housekeeping genes to determine the identities of potential A. rivipollensis and A. media species isolates also used in this study. In the process of species identification, the isolate Aeromonas sp. 5717 by 16S rRNA sequence clustered together in the NJ (neighbor-joining) tree with A. media, but in the GyrB NJ tree this isolate clustered with A. rivipollensis. The better practice would be identifying Aeromonas species by concatenated sequences of several housekeeping genes or by WGS (Lee et al., 2023). Although WGS would be a significant upgrade in terms of reliability, we are confident that the identification by the methods used in this study is sufficient to draw the conclusions to the degree presented in line with our results (and limitations of the methodology used).
Some papers on the topic in case of advanced interest in the subject:
- Küpfer, M., Kuhnert, P., Korczak, B. M., Peduzzi, R., & Demarta, A. (2006). Genetic relationships of Aeromonas strains inferred from 16S rRNA, gyrB and rpoB gene sequences. International Journal of Systematic and Evolutionary Microbiology, 56(12), 2743–2751. https://doi.org/10.1099/ijs.0.63650-0
- Persson, S., Al-Shuweli, S., Yapici, S., Jensen, J. N., & Olsen, K. E. P. (2015). Identification of Clinical Aeromonas Species by rpoB and gyrB Sequencing and Development of a Multiplex PCR Method for Detection of Aeromonas hydrophila, A. caviae, A. veronii, and A. media. Journal of Clinical Microbiology, 53(2), 653–656. https://doi.org/10.1128/JCM.01963-14
- Fernández-Bravo, A., & Figueras, M. J. (2020). An Update on the Genus Aeromonas: Taxonomy, Epidemiology, and Pathogenicity. Microorganisms, 8(1), Article 1. https://doi.org/10.3390/microorganisms8010129
- Navarro, A., & Martínez-Murcia, A. (2018a). Phylogenetic analyses of the genus Aeromonas based on housekeeping gene sequencing and its influence on systematics. Journal of Applied Microbiology, 125(3), 622–631. https://doi.org/10.1111/jam.13887
- Talagrand-Reboul, E., Roger, F., Kimper, J.-L., Colston, S. M., Graf, J., Latif-Eugenín, F., Figueras, M. J., Petit, F., Marchandin, H., Jumas-Bilak, E., & Lamy, B. (2017). Delineation of Taxonomic Species within Complex of Species: Aeromonas media and Related Species as a Test Case. Frontiers in Microbiology, 8. https://www.frontiersin.org/journals/microbiology/articles/10.3389/fmicb.2017.00621
- Lee, H.-J., Storesund, J. E., Lunestad, B.-T., Hoel, S., Lerfall, J., & Jakobsen, A. N. (2023). Whole genome sequence analysis of Aeromonas spp. isolated from ready-to-eat seafood: Antimicrobial resistance and virulence factors. Frontiers in Microbiology, 14. https://www.frontiersin.org/journals/microbiology/articles/10.3389/fmicb.2023.1175304
Reviewer 1: Another unclear point is how other Aeromonas strains were taxonomically identified (Table 1)? Was it 16SpPHK sequencing or some other method? Please, clarify this.
Author response: As these strains were isolated in the span of several years (e.g., a temporal difference of 7 years between the isolation of some of them), the methodology for their initial identification before storage was also different and was not unified. This was one of the reasons why re-identification/validation of their identity using Sanger-based sequencing of 16S rRNA and gyrB gene was carried out within this work (elaborated on previously in answer to the commentary regarding section 2.8.). The species designations in Table 1 currently are in line with the phylogenetic analysis of the first 16S rRNA (which was not very efficient for the delineation of several closely related Aeromonas species, and also, for example, for A. media – multiple 16S rRNA gene copies seem to differ between themselves quite a lot), and then gyrB to improve the resolution within the context of different Aeromonas species isolated elsewhere.
The rationale is mentioned in Lines 310-313 with an appropriate reference for a follow-up reading for those who may become interested:
“As 16S rRNA sequence phylogenies are known to be inadequate in discerning between some of the currently recognized closely related Aeromonas species, the resolution of the available Aeromonas strain identification was improved with the gyrB gene sequence analysis of the isolates [54].”
Reviewer 1: Technical comments: Abstract, Line 15. “genomic analysis indicates” not “indicate”.
Author response: Although we have initially used “Genomic analyses indicate …” (the plural form) as different types of analysis were performed, we see that all of them can be perceived holistically as a single “genomic analysis”. The corresponding line has been revised accordingly and now begins with “Genomic analysis indicates …”.
Reviewer 1: Section 2.9. Lines 340-343 are reduntant. May be shortened.
Author response: The corresponding shortened sentence now reads: “The corresponding shortened sentence now reads: “Phage genomic DNA was extracted from 300 µL of CsCl gradient-purified and desalted JELG-KS1 specimen employing the same protocol utilized for bacterial genomic DNA extraction (section 2.1.).”
Reviewer 1: Section 3.1. Lines 448-450. Please, clarify or re-write the phrase “sufficiently broad range of possible plaque morphologies”.
Author response:
We have used this phrase to mention that the phage JELG-KS1 showed various plaque size ranges and appearances (e.g., clear or turbid) depending on the Petri dish incubation temperature.
We are sorry if this could have been misinterpreted. The wording of the appropriate paragraph in the revised version of the manuscript is now slightly changed to, hopefully, be more straightforward, this paragraph now reads:
“On the lawn of its isolation host Aeromonas salmonicida JK, phage JELG-KS1 demonstrated a range of possible plaque sizes and appearances in the double agar overlay as-says with 0.7% LB as the top medium (Fig 1). Phage JELG-KS1 plaque appearance as well as possible plaque diameter range was found to be dependent on the double agar overlay assay plate incubation temperature, although the efficiency of plating (observed plaque count) was not differing between double agar overlay assays incubated overnight at RT, 30 °C, and 37 °C (Kruskal-Wallis test, p = 0.34). After incubation at RT and 30 °C plaques were clear and large, whereas on a plate incubated at 37 °C plaques were small, with some of them demonstrating remarkable turbidity.”
Reviewer 1: Materials and Methods, Line 170. Please, add the manufacturer of the orbital shaker.
Author response: The shaker we used was an Infors Multitron incubator shaker (Infors AG, Bottmingen, Switzerland). This has now been added to the relevant line of the manuscript.
Reviewer 2 Report
Comments and Suggestions for Authors
The manuscript entitled “Isolation and characterization of novel Aeromonas salmonicida-infecting Studiervirinae bacteriophage JELG-KS1” is an interesting study. Some details are recommended for the authors:
1. The authors are recommended to include information about the selection of the sampling site, specifically why they chose that site if the search is for bacteriophages that control diseases in fish.
2. Additionally, did they work with a non-pathogenic strain of Aeromonas salmonicida in fish?
3. Lastly, the authors suggest that they might need to follow a different sampling method. Could they briefly include how they would do this? For example, how would they search for a bacteriophage that infects various species of pathogenic fish bacteria? Would they include other types of sampling sites?
Author Response
Reviewer 2
The manuscript entitled “Isolation and characterization of novel Aeromonas salmonicida-infecting Studiervirinae bacteriophage JELG-KS1” is an interesting study. Some details are recommended for the authors:
Author response: We thank Reviewer 2 for the initiation of an interesting discussion with his/her review and the opportunity to tell more about what we are up to in our laboratory more broadly outside the scope of this particular paper. We hope that our response makes it clearer in terms of the rationale behind this study and how the “random” isolation of a phage having potentially worthwhile properties on a bacterial strain representing a species of importance kickstarts a new research direction in our lab. We have additionally tried to read through the manuscript once more to fix small writing mistakes and typos we could spot. We look forward to the second round of revisions in case of necessity!
Reviewer 2: The authors are recommended to include information about the selection of the sampling site, specifically why they chose that site if the search is for bacteriophages that control diseases in fish.
Author response: The isolation of A. salmonicida strain JK and phage JELG-KS1 is actually one of the results that stemmed from retrieving whatever environmental bacterial strains and their phages from wastewater to expand the known bacteriophage diversity. However, the promising infection properties this phage demonstrated on its isolation host strain (that turned out to be a mesophilic strain of A. salmonicida) prompted the idea of testing its ability to infect various relevant Aeromonas isolates previously retrieved from local aquaculture environments and subsequently stored in the BIOR collection (specifically isolated from infected fish, as Aeromonas is generally thought of as a fish pathogen). However, as is demonstrated within this work, it was found that the host range of this phage is limited to the original isolation strain and neither of the strains directly relevant to the aquaculture settings were found to be susceptible to infection by JELG-KS1.
Wastewater is one of the most popular sources to look for novel bacteriophages infecting a wide variety of hosts, and phages able to productively infect a broad range of different bacterial genera representatives can be readily retrieved. We agree that it seems kind of obvious that for retrieval of phages relevant to biocontrol, one should try to use relevant strains as the isolation hosts/indicator cultures in the first place. However, reports of phages that can productively infect both environmental and epidemiologically relevant strains of bacteria, but were initially isolated from wastewater are not that rare. The initial goal of this work, however, was training in phage-related techniques for one of the new members of our lab, and isolation of JELG-KS1 seemed what might be a fortunate turn of events prompting us to test its potential application possibilities for biocontrol of Aeromonas spp.
Reviewer 2: Additionally, did they work with a non-pathogenic strain of Aeromonas salmonicida in fish?
Author response: Unfortunately, we are not sure about the pathogenicity of A. salmonicida strain JK. As many reports of the association between Aeromonas spp. and diseases of various animals (including humans) could be found in the literature, the experimentation was carried out according to the possible risks of the strain JK being pathogen (PPE worn at all times when handling the strain, regular decontaminations of work surfaces, etc.). The other 19 Aeromonas spp. strains that the JELG-KS1 phage was challenged against were all isolated from diseased fish (assumed to be the etiological agents of the respective disease symptoms, and, thus, pathogenic).
Reviewer 2: Lastly, the authors suggest that they might need to follow a different sampling method. Could they briefly include how they would do this? For example, how would they search for a bacteriophage that infects various species of pathogenic fish bacteria? Would they include other types of sampling sites?
Author response: Thank you for this question! As is indicated in the first point of our response, particularly this study was initially conducted without the intention of specifically searching for fish pathogens or fish pathogen-infecting phages. However, the failure of JELG-KS1 to lyse the relevant strains pathogenic to fish inspired us to try securing funding for a project aimed at specifically hunting for Aeromonas phages that are able to efficiently infect Aeromonas spp. strains from local aquaculture associated with the disease in fish. Thus, this particular study has actually kickstarted a project where we are now intentionally isolating phages for Aeromonas spp. known to be pathogenic to fish. Within the project we are currently implementing, we are working with both the Aeromonas BIOR strains listed within this manuscript that phage JELG-KS1 failed to lyse, as well as other contemporary strains corresponding to the most recent Aeromonosis outbreaks at several Latvian fish farms. Not surprisingly, a lot of phages that are able to infect these “relevant” strains can also be consistently retrieved from wastewater samples of different Latvian cities, not only from the infected fish tissue samples and relevant tank waters. Preliminary data we currently have demonstrates that with this sampling technique (from the ~30 genetically different Aeromonas phages that we have isolated in our lab up to now within this project) we can formulate several combinations of phages for a “cocktail” that would be able to efficiently lyse majority of the strains that JELG-KS1 failed to lyse (even exclusively from the phages isolated from wastewater, but isolated on relevant strains). Thus, we would once again highlight that the sample source is not a bottleneck for isolation of phages capable of productively infecting Aeromonas strains isolated from aquaculture, but usage of such strains as initial indicator cultures ensures the relevance of the phages being isolated (contrary to the case of JELG-KS1 which was isolated on an “environmental” strain of Aeromonas salmonicida, although, given the phage diversity out there this might just be a very unfortunate turn of events that it seems to be so specific). Even from the same sample one can find several distinct phages infecting different representatives of the same bacterial genera, and we believe that having a well-characterized panel of epidemiologically relevant host strains (including the whole genome sequences, which we are also hoping to elucidate in the near future, to aid in appropriate host strain selection for inclusion in such panel) would even further improve such efforts by allowing to choose strains that are more diverse from each other for phage hunting (potentially leading to a wider diversity of phages being retrieved).
Although the things described above are a work largely still ongoing, we are optimistic that we might present a fully-fledged paper or even a series of papers detailing the proceedings of this project starting at the end of this year. Aeromonas spp. and their phages seem to be fascinating enough for us to work with and we hope to contribute to this line of research more ourselves in the near future. The main principle that guides our further phage hunting is already mentioned in the manuscript “… isolation of novel bacteriophages for a potential practical application in biocontrol should better be carried out using relevant host strains associated with the setting in which the biocontrol application is ought to eventually take place to reduce the risk of isolating a priori irrelevant highly host-specific phages associated with irrelevant host strains.”.
Reviewer 3 Report
Comments and Suggestions for Authors
Comments and suggestions
Microorganisms-2889182
Abstract section.
1. The results part in this section should be clarified.
2. Indicate some examples of fish species.
Keywords:
3. A maximum of 5 keywords would be better. They must all be MeSH terms.
Introduction section:
4. The concepts of podophage and phage biocontrol should be mentioned.
Materials and methods section.
5. Reorganize this section, perhaps a work algorithm will help to organize the processes.
6. Reference 39 is relevant?
Results section:
7. Figure 3 is not clearly seen, check if it meets the image resolution characteristics requested by the magazine.
Discussion section:
8. Highlight the limitations of the work based on the techniques used.
Conclusions section:
9. Conclusions must be based on the objectives of the research
Comments on the Quality of English LanguageMinor editing of English language required
Author Response
Reviewer 3
Author response: We thank Reviewer 3 for pointing out several things that were initially overlooked! Please, find our replies and explanations below. We have additionally tried to read through the manuscript once more to fix small writing mistakes and typos we could spot. Should any of the points remain unaddressed properly in your opinion during this round of revisions, we are ready to resolve the remaining issues during the next revision round.
Abstract section.
Reviewer 3: 1. The results part in this section should be clarified.
Author response: We have attempted to write the abstract as per the instructions for the authors section of the MDPI Microorganisms journal with a particular emphasis on the results, as we see it. Given the fact that results currently already represent ~ 2/3 of the abstract (that is limited to 200 words as per the specifications, and currently is precisely 200 words in length), but it also has to include some background, etc., we see no real possibility to elaborate on the results in the abstract more.
Reviewer 3: 2. Indicate some examples of fish species.
Author response: The main focus of this study is the retrieval of novel phage JELG-KS1 on a strain of A. salmonicida from wastewater. As this phage failed to lyse strains representing other Aeromonas species, as well as even several other A. salmonicida strains retrieved from fish, we do not think it is worth mentioning particular fish species in the abstract so as to not give a false impression of the possible relevance of JELG-KS1 to the treatment of Aeromonosis in them.
The fish species from which the Aeromonas isolates JELG-KS1 failed to lyse were retrieved and are mentioned in Table 1 of the Materials and Methods section. As you see, there is quite a variety even when talking about just several Aeromonas spp. we have worked with, squeezing in the relevant fish species names Aeromonas tend to infect into the abstract would also not be objectively possible due to the word count limitation.
Keywords:
Reviewer 3: 3. A maximum of 5 keywords would be better. They must all be MeSH terms.
Author response: We thank Reviewer 3 for this suggestion, we have not actually thought about search engine optimization and just used keywords that we thought were representative of the content, without paying attention to whether they were MeSH terms.
Although it seems that MDPI Microorganisms allows up to ten pertinent keywords without explicitly asking for them to be MeSH terms, most of the initial keywords were dropped in the revised version as per your request.
Currently, the revised keywords (which are all MeSH terms) are:
Bacteriophage, whole genome sequencing, genomics, aquaculture, Aeromonas salmonicida
Introduction section:
Reviewer 3: The concepts of podophage and phage biocontrol should be mentioned.
Author response:
Although both terms are very common in the field of phage research, we agree that part of the potential readership of this special issue might not be from the field, thus, we have opted to include elaborations on the meaning behind the concept of podophage.
The first two sentences of the last paragraph of the introduction now include an explanation about the “podophage” concept and read: “In this study, a novel lytic bacteriophage JELG-KS1 infecting a strain of Aeromonas salmonicida retrieved from wastewater is reported and thoroughly characterized. Transmission electron microscopy revealed that JELG-KS1 virions have a podophage morphotype (icosahedral capsid with a short non-contractile tail).”
As for the biocontrol concept, we believe that, actually, lines 67-86 already explain the concept of biocontrol, especially specifically using phages. And it would, as we see it, be inappropriate to mention that the “bio” part in the biocontrol points to “methods of controlling pests, pathogens, and other harmful organisms using biological objects or their derivatives” (or something along these lines) in a scientific journal, as the concept is rather simple and cross-field, it probably should be known to all the potential readerships.
If you, however, insist – we can add such an explanation of biocontrol to the relevant paragraph during the second round of revisions.
Materials and methods section.
Reviewer 3: 5. Reorganize this section, perhaps a work algorithm will help to organize the processes.
Author response: We are not sure what was meant by the respected Reviewer 3. The current sequence of methods section subheadings actually sequentially represent the steps taken during this study in the order they happened. If this was meant by the “work algorithm” to organize the processes, then this is presented precisely as it happened (some elaboration on this can also be read in responses to Reviewers 1 and 2). This stems from the fact that the scope of what was initially planned got expanded based on the promising results we got when characterizing the properties of JELG-KS1 on its isolation host A. salmonicida JK.
Reviewer 3: 6. Reference 39 is relevant?
Author response: It depends on the subjective preference of a particular researcher, in our experience, quite a lot of the reviewers that we encountered previously have explicitly asked for the references to even for some of the most well-known protocols of the field to be mentioned (this particular one included). Although reference 39 (introduction of Sanger sequencing by F. Sanger and colleagues, basically) is, indeed, arguably, one of the most important and known papers in molecular biology (e.g., cited in 87,766 different documents at the time of writing this response) – it still is actually directly relevant to several experiments performed within our work and their results. We would like to retain this reference.
Results section:
Reviewer 3: 7. Figure 3 is not clearly seen, check if it meets the image resolution characteristics requested by the magazine.
Author response: Thank you for bringing this to our attention. Figure 3 was added as part of the submission as a 5794 x 3181 px file with a resolution of 600 dpi. As the journal is online-only, we believe that this quality will be sufficient to allow zooming in to the image for inspection even of the details that might not be sufficiently clear at 100% size on an A4 paper. Everything looks sharp in the original file, and we really hope that the editorial office or someone from the journal team might help to address this, if there is any issue with the image during conversion that results in a loss of quality within the manuscript itself.
Discussion section:
Reviewer 3: 8. Highlight the limitations of the work based on the techniques used.
Author response:
Well, the main limitation is the fact that the host range of this phage turned out to seemingly be so narrow - a recurring theme throughout this manuscript. Thus, we could not investigate how the infectious properties (e.g., EOP, burst size, life cycle characteristics, plaque size, and morphology) might have differed among strains/species of its host, as it was only able to infect its isolation host. There certainly was the available number of Aeromonas spp. limitation, specifically A. salmonicida strains, and more strains might have been used (already mentioned in lines 538-541), but, we believe, the number and diversity of the spatio-temporally different Aeromonas strains used is enough to conclude that JELG-KS1 is not a candidate expected to be helpful for phage-based biocontrol applications in local aquaculture. Another limiting factor could be the composition of the available host range testing panel based on their properties, particularly given the lack of other mesophilic Aeromonas salmonicida strains available for testing. We have identified our strains using 16S rRNA and gyrB sequences, but it would be way better to have strains used for host range screens characterized by WGS (on this we are starting to work on, but the problem is broader due to the fact that genomic epidemiology of Aeromonas spp. was not previously carried out in Latvia to the best of our knowledge, so we have to start “from scratch”). Please see elaborate answers to Reviewer 2 points on how we are trying to approach studies of Aeromonas spp. and their phages thanks to the isolation of JELG-KS1 that inspired us to start going further in this research direction. Although there is always room for improvement (more experimental conditions, different twists on experimental designs, etc.), we think that the techniques used for phage characterization presented herein should be in line with what is at the core of current state-of-the-art, especially the in silico analyses.
Conclusions section:
Reviewer 3: 9. Conclusions must be based on the objectives of the research
Author response: This is a tricky one. There were no actual specific objectives when this particular research started, as the main goal was to perform a “hypothesis-free” and culture-based prospecting for interesting phages that would be sufficiently divergent from the ones isolated and characterized so far. The favorable infectious properties of this particular phage that was rather intergenomically different from all the other phages sequenced so far up to date prompted us to explore its ability to be used for biocontrol of Aeromonas spp. relevant for aquaculture, and, as can be seen from the results – JELG-KS1 turned out to be clearly not a good phage candidate for such purposes due to its host range specificity. Thus, we believe that Conclusions are currently highlighting the key findings of our study sufficiently well without reciting the results presented in the other parts of the manuscript and underscore the key takeaway from this paper - the significance of rigorously testing isolated bacteriophages before proposing their potential role in biocontrol of infectious diseases.
Round 2
Reviewer 3 Report
Comments and Suggestions for Authors
I am satisfied with the authors' responses. I have no more comments or suggestions
Comments on the Quality of English LanguageMinor editing of English language required